# The Use of Non-Invasive Continuous Blood Pressure Measuring (ClearSight^®^) during Central Neuraxial Anaesthesia for Caesarean Section—A Retrospective Validation Study

**DOI:** 10.3390/jcm11154498

**Published:** 2022-08-02

**Authors:** Philipp Helmer, Daniel Helf, Michael Sammeth, Bernd Winkler, Sebastian Hottenrott, Patrick Meybohm, Peter Kranke

**Affiliations:** 1Department of Anaesthesiology, Intensive Care, Emergency and Pain Medicine, University Hospital Würzburg, Oberdürrbacher Str. 6, 97080 Würzburg, Germany; helmer_p@ukw.de (P.H.); helf_d@ukw.de (D.H.); sammeth_m@ukw.de (M.S.); bernd.e.winkler@gmail.com (B.W.); hottenrott_s@ukw.de (S.H.); meybohm_p@ukw.de (P.M.); 2Department of Applied Sciences, Coburg University, Friedrich-Streib-Str. 2, 96450 Coburg, Germany

**Keywords:** volume clamp, epidural anaesthesia, supine hypotensive syndrome, ClearSight^®^, Bland–Altman, Fisher Z-score transformation, Pearson correlation coefficient

## Abstract

The close monitoring of blood pressure during a caesarean section performed under central neuraxial anaesthesia should be the standard of safe anaesthesia. As classical oscillometric and invasive blood pressure measuring have intrinsic disadvantages, we investigated a novel, non-invasive technique for continuous blood pressure measuring. **Methods:** In this monocentric, retrospective data analysis, the reliability of continuous non-invasive blood pressure measuring using ClearSight^®^ (Edwards Lifesciences Corporation) is validated in 31 women undergoing central neuraxial anaesthesia for caesarean section. In addition, patients and professionals evaluated ClearSight^®^ through questioning. **Results:** 139 measurements from 11 patients were included in the final analysis. Employing Bland–Altman analyses, we identified a bias of −10.8 mmHg for systolic, of −0.45 mmHg for diastolic and of +0.68 mmHg for mean arterial blood pressure measurements. Pooling all paired measurements resulted in a Pearson correlation coefficient of 0.7 for systolic, of 0.67 for diastolic and of 0.75 for mean arterial blood pressure. Compensating the interindividual differences in linear regressions of the paired measurements provided improved correlation coefficients of 0.73 for systolic, of 0.9 for diastolic and of 0.89 for mean arterial blood pressure measurements. **Discussion:** Diastolic and mean arterial blood pressure are within an acceptable range of deviation from the reference method, according to the Association for the Advancement of Medical Instrumentation (AAMI) in the patient collective under study. Both patients and professionals prefer ClearSight^®^ to oscillometric blood pressure measurement in regard of comfort and handling.

## 1. Introduction

The incidence of hypotension during the induction of central neuraxial anaesthesia for caesarean section is up to 74% [1], resulting in a relevant sequelae in obstetric anaesthesia [2]. Despite the physiological adaptations of the cardiovascular system in pregnancy, the reduction in total peripheral resistance and the impaired venous return, among other factors, lead to an increased risk for pregnant women developing hypotension, especially in conjunction with spinal anaesthesia [3,4,5]. Hypotension is associated with adverse effects for mother and child, including foetal acidosis as well as nausea, vomiting, dizziness and hypoxaemia in the mother to be [6]. The use of uterotonic drugs, such as oxytocin or carbetocin, further reduce vascular resistance and hypotensive episodes may also occur in the course of surgery. Because of possible negative effects, supine hypotension syndrome must also be diagnosed and treated rapidly [7]. These facts highlight the need for possibilities to closely monitor blood pressure during obstetric anaesthesia while minimising the discomfort for the patient, who usually is awake during the procedure [4]. 

In the perioperative setting, blood pressure usually is measured oscillometrically by pneumatic transmission between the brachial artery and an upper arm cuff. The mean arterial pressure (MAP) is measured directly, whereas the systolic and the diastolic arterial pressures are derived using corresponding algorithms [8]. MAP is defined as the lowest cuff pressure with the highest oscillation in the arterial curve [9]. However, the literature reports that such an approach may lead to an underestimation of high arterial blood pressures and also to an overestimation of low blood pressures [10]. In addition, there are subtle differences across manufactures in the algorithm employed to calculate the systolic and diastolic blood pressure values [8]. In fact, Bur et al. concluded that cuff-based measurements of blood pressure are inappropriate for use in intensive care patients [11]. In addition to problems related to mis-cuffing (incorrect size ratio of the upper arm to cuff), many conscious patients consider the measurements uncomfortable, particularly because the time intervals between the single measurements are usually kept short in order to quickly detect hypotensive episodes. 

The gold standard for measuring blood pressure in an operating room or intensive care unit (ICU) still relies on invasive measurements through the catheterisation of an arterial vessel. The key advantage is the direct beat-to-beat measurement of the blood pressure. Maheshwari et al. demonstrated that during non-cardiac surgery the continuous monitoring of haemodynamic parameter is able to significantly lower the rate of hypotension compared to non-continuous monitoring [12]. However, measuring with an arterial catheter can cause various complications, such as infections, haematomas, tissue and nerve damage and thromboembolic events. Therefore, a routine invasive measurement of blood pressure for caesarean sections in healthy pregnant women is inappropriate.

ClearSight^®^ (Edwards Lifesciences Corp., Irvine, CA, USA), has been conceived to complement the advantages of both invasive and non-invasive blood pressure measuring. This device non-invasively outputs a continuous blood pressure curve. To investigate the accuracy of ClearSight^®^ for blood pressure monitoring during labour, we compared ClearSight^®^ against the established oscillometric method during an extended evaluation-phase and asked the patients and attending physicians regarding their preferences for distinct methods to measure the blood pressure. These data were obtained during clinical routine and were retrospectively analysed and described in this manuscript.

## 2. Materials and Methods

Within the context of quality management of an extended non-invasive haemodynamic monitoring system in obstetric anaesthesia, vital parameters were recorded simultaneously according to both established clinical protocols (Philips MX750 and Philips IntelliVue X3) and ClearSight^®^ (Edwards Lifesciences Corp.). Measurements were obtained between January 2021 and March 2021 at the Department of Anaesthesiology, Intensive Care, Emergency and Pain Medicine at the University Hospital Würzburg. The data were analysed retrospectively in accordance with Article §27 of Bavarian hospital law as they were obtained during routine procedures. In addition, we declared all analyses performed on these data to the local ethics committee of Würzburg (ref. nr. 20210903/01). For our benchmark of ClearSight^®^ against oscillometric blood pressure measurements, we only included patients undergoing primary or secondary, non-emergency caesarean sections. Epidurals, spinal or combined spinal epidural anaesthesia (CSE), were required for inclusion. Exclusively patients with sinus rhythms and more than five data points for comparison between the two measuring methods were considered in our analysis. The primary endpoint was defined as the difference between oscillometric blood pressure measurement and ClearSight^®^ according to the Association for the Advancement of Medical Instrumentation (AAMI). Secondary endpoints were defined as patient comfort and physician’s preferences for blood pressure measuring methods.

After implementing clinical monitoring and ClearSight^®^, single-shot spinal anaesthesia (SSA) was performed in a sitting position under sterile conditions according to established clinical protocols. Parallel to co-loading with 1000 mL of balanced crystalloid infusion solution (Sterofundin©), lumbar spinal anaesthesia (level: L3/4 or L4/5) was performed with an atraumatic Sprotte needle (24G Sprotte©, FA Pajunk^®^, Geisingen, Germany). After the positive aspiration of cerebrospinal fluid (CSF), 9–10 mL of a mixture containing local anaesthetic and opioid (4 ml bupivacaine 0.25% isobaric plus sufentanil 5 µg plus 5 mL Sodium chloride 0.9%) were carefully injected intrathecally [13]. Epidural anaesthesia and CSE were performed following, in principle, the SSA protocols, but employing a Touhy needle for the epidural puncture (18G, FA Pajunk^®^) and, respectively, a special CSE needle (18G epidural and 25G spinal needle, EpiSpin Safety©, FA Pajunk^®^). A loss-of-resistance to the saline method was used to identify epidural space, and subsequently a catheter (EpiLong Standard©, FA Pajunk^®^) was inserted epidurally. In case of insufficient analgesia, ropivacaine 0.75% (up to 10 mL) was administered. If an epidural catheter was already in place, a top-up with 15–20 mL of ropivacaine 0.75% was administered for secondary caesarean section. Adequate analgesia was confirmed when a TH5 level was achieved. Hypotension was treated according to established clinical protocols, at the discretion of the attending anaesthesiologist. Hypotension was defined by clinical standards as MAP < 65 mmHg or a systolic blood pressure < 90 mmHg. 

In addition to standard monitoring by cuff-based, non-invasive blood pressure measurements (NIBP) on the right upper arm, a ClearSight^®^ finger sensor was attached to the left index finger. Concurrently, vital and haemodynamic parameters were displayed in parallel on a monitor (Philips MX750). These finger sensors were available in three different sizes that could be further adapted individually. Data were stored in the anaesthesia database of the hospital’s patient data management system (PDMS), Copra (COPRA System GmbH, Berlin, Germany, V6.84.2). As part of the standard electronic documentation system in the operating room, body-mass-index (BMI), physical status according to ASA (American society of Anesthesiologists) classification, type of intervention, the medication administered and the measurement location of ClearSight^®^ were routinely documented. Both the anaesthesiologists and the patients were surveyed on a voluntary basis regarding their preferences for either of the employed measuring devices. Corresponding results served as internal quality management and provided practical experiences when deciding on acquiring the ClearSight^®^. 

### Statistical Analysis

All statistical analyses of the primary data were performed employing the R platform (V.4.1.0). Following reports from the literature and recommendations by AAMI, we considered a difference of 5 mmHg with a SD of ±8 mmHg between the compared measurement methods as acceptable [14]. To investigate the presence of systematic biases, we employed Bland–Altman plots, subtracting ClearSight^®^ blood pressure measurements from the corresponding values obtained by the reference method [14,15]. We first performed linear regression and calculated the Pearson Correlation Coefficient (PCC) separately on the data points of each of the patients. Subsequently, we employed two different approaches to obtain a combined PCC value: (i) adopting common proceeding, we pooled all available data points from different patients and analysed them altogether. However, pooling data points with possibly different linear correlations—and consequently also different Pearson coefficients—leads to an underestimation of the true correlation. Therefore, we also employed (ii) an established method to combine the PCC values computed individually for each patient, based on the weighted average of intermediately transformed Z-scores [16]. In order to confirm the statistical power of our sample size, we employed the function r.test() from the R package “pwr”: for the minimal observed PCC of 0.73, a power of 0.8 and a significance level of 0.05, and we thus obtained a minimal sample size of *n = 11*.

## 3. Results

In total, ClearSight^®^ was used in 31 subsequent childbearing patients undergoing caesarean section. Twenty-seven patients underwent primary and four patients underwent secondary caesarean section. From the beginning, we excluded in our analysis two patients who underwent caesarean sections under general anaesthesia. Spinal anaesthesia was established in twenty patients, eight patients received CSE and in one patient epidural anaesthesia was topped up for a secondary section. A patient with atrial fibrillation was further excluded from the analysis. Seventeen patients could not be included in the analysis due to insufficient data quantity with less than five paired measurements, because non-invasive blood pressure measuring was deactivated by the attending anaesthesiologist after establishing ClearSight^®^. The period during which measurements were observed thereby extended over the entire intervention period with a measurement interval of the non-invasive blood pressure of >5 min. Eventually, eleven patients (eight patients with spinal anaesthesia and three patients with CSE) with, in total, 139 paired measurements, were included in our further analysis (Table 1). 

ClearSight^®^ measurements were collected from the left index finger for 96.7% of the patients (*n = 31*). Almost all patients developed hypotension, requiring vasopressor therapy during the placement of neuraxial anaesthesia, including the subsequent caesarean section. Of the patients, 96.7% received cafedrine/theodrenaline (Akrinor^®^) to treat hypotension. A satisfaction survey showed that 93.4% of the patients and 96.7% of the anaesthesiologists preferred ClearSight^®^ to conventional oscillometric measurement. Table 2 summarizes that the average age of the patients was 36 years, with an average BMI of 34.2 kg/m^2^ and ASA category II.

We employed two different statistical approaches to validate the accuracy of the measurements. In our first approach, we pooled the paired measurements from all patients segregated by the type of blood pressure measurement (i.e., MAP, systolic or diastolic blood pressure) and first assessed collinearity by linear regression and calculating the corresponding PCC. Compared to the oscillometric measurement, ClearSight^®^ achieved PCC values of 0.67 (95% CI 0.56 to 0.75; *p* < 0.001) for diastolic blood pressure, a PCC of 0.7 (95% CI 0.6 to 0.77; *p* < 0.001) for systolic blood pressure and a PCC of 0.75 (95% CI 0.67 to 0.81; *p* < 0.001) for mean blood pressure (Figure 1). 

Figure 1 also shows the linear models regressed on the paired measurements of each blood pressure measurement. It is of note that the slopes and the shifts of linear models, which regressed distinctly on the paired measurements of each patient, differ from the pooled model (Figure 1) and also from each other (Figure 2). These observations suggest that the correlation between the compared measuring methods is subject to individual factors intrinsic to each treatment, and consequently indicate that assessing the linear correlation of the paired measurements pooled from different patients results in distorted results. By definition, PCC values are non-additive and therefore also cannot be straightforwardly combined, e.g., by calculating a (weighted) average value. We therefore sought to obtain an unbiased PCC value for all patients in our study by computing the weighted average of correspondingly transformed Z-scores of the individual PCCs, which in turn could be transformed back into a PCC value. Following this approach, we achieve substantially higher PCC values than in our first approach on pooled measurements: 0.9 for diastolic blood pressure, 0.73 for systolic blood pressure and 0.89 for mean blood pressure.

Subsequently, we also assessed the presence of systematic biases between the two investigated measuring methods employing Bland–Altman plots. Our analyses pinpoint a bias value of −0.45 mmHg for diastolic blood pressure, a bias of −10.8 mmHg for systolic blood pressure and a bias of +0.68 mmHg for mean blood pressure (Table 3, Figure 3).

## 4. Discussion

The results of our pilot study demonstrates that ClearSight^®^ can be employed for continuous, non-invasive blood pressure monitoring during caesarean sections under neuraxial anaesthesia. Based on our small cohort, an acceptable agreement, according to the AAMI, between the established oscillometric blood pressure measurement with ClearSight^®^ for diastolic and mean arterial pressure was shown. However, systolic blood pressure measurements are beyond these limits of acceptance. Based on a survey of professionals and patients, ClearSight^®^ is superior to the conventional oscillometric blood pressure measuring both concerning early warnings for hypotensive episodes and also with respect to better patient comfort.

Blood pressure measurements by ClearSight^®^ are based on two distinct methods, namely the volume clamp method and the so-called “PhysioCal”. Based on the volume clamp method, the diameter of the artery is kept constant by applying external pneumatic pressure by a finger cuff [17]. This diameter is monitored by photoplethysmography [18]. Adjustments of the counter-pressure of the finger cuff are measured with high frequency (approx. 1000 Hz) and provided as parameters to the algorithm calculating blood pressure. The “PhysioCal” method is used to calibrate the system, and subsequently blood pressure measurements of the finger artery are employed to reconstruct the arterial pressure of the brachial artery as an established parameter. Based on pulse contour analysis, besides blood pressure and the heart rate, stroke volume, stroke volume variation, systemic vascular resistance and cardiac output can also be calculated [19]. 

Table 4 provides an overview of selected studies, which further validate the measuring accuracy of the ClearSight^®^ in different cohorts. Our observations that the deviation of systolic blood pressure measurements by ClearSight^®^ are outside the AAMI limits confirm the results of the study by Tani et al., Rogge et al. and Schumann et al. [14,20,21]. The largest cohort was studied by Schumann et al. (*n* = 90) and Rogge et al., with the most data points of almost 100,000 [14,21]. Furthermore, Takashi et al. showed the reduced incidence of hypotension during caesarean sections for ClearSight^®^ compared to the oscillometric control group [22]. In addition to monitoring real-time blood pressure, ClearSight^®^ offers the ability to predict hypotension based on a machine-learning index—the so-called “Hypotension Prediction Index” (HPI). Frassanito et al. determined a sensitivity and also a specificity of 0.85 for ClearSight^®^ by correctly predicting hypotension 15 min before the event under general anaesthesia in major surgical procedures [23]. Focusing on awake caesarean sections, Frassanito et al. demonstrated a sensitivity and specificity of 0.97 in HPI prediction accuracy 2 min before the onset of hypotension [24]. 

Haemodynamic parameters, such as stroke volume, stroke volume variation, systemic vascular resistance, and cardiac output, play an important role, particularly in pregnant women due to their physiological changes. Duclos et al. therefore compared the accuracy of stroke volume and cardiac output measurements between ClearSight^®^ and the established standard (transthoracic echocardiography) in 44 pregnant women [30]. The authors concluded that the observed results were not within an acceptable range in the patient population under study. However, the incidence of nausea during caesarean section could be reduced by ClearSight^®^-guided goal-directed fluid therapy [31]. Also according to Lee et al., ClearSight^®^ is a valid alternative, especially for surgical procedures with limited access to the patient’s upper arms [25]. Highlighting patient comfort, Eley et al. showed in 450 pregnant women in their third trimester that the finger cuff of ClearSight^®^ can be placed easily [32].

Besides the investigated volume-clamp method, other alternative methods for blood pressure measurements are pulse wave velocity/pulse transit time or through applanation tonometry [33,34]. Furthermore, methods for non-invasive blood pressure measurement based on photoplethysmography are also currently being investigated, especially in the sector of wearables or fitness trackers [35]. With pulse transit time measurements, blood pressure is calculated from the time latency between electrical activity and mechanical peripheral pulse wave, employing a pre-calibrated system. In contrast, applanation tonometry requires external pressure compresses of an artery against an abutment (e.g., a bone) but without occlusion. The blood pressure is then obtained by direct measurements of a device located in immediate vicinity to the artery. However, as these methods are currently very susceptible to artifcats, they are not recommended for the use in clinical routine.

Regarding the interpretation of our results, some constraints may arise due to the retrospective and single-centre design of our study, allowing us to investigate only a limited number of cases. Due to the limited sample size, we cannot confidently exclude the possibility of under powering some of the single patient analyses. However, the number of paired measurements allowed the conclusion of a significant correlation between ClearSight^®^ and the reference method. Importantly, we demonstrate that in our dataset the linear models, which regressed for the paired measurements of each patient individually, differ in slope and shift. Hence, a straightforward analysis on measurements obtained from different patients underestimates the true correlation, but biases can successfully be alleviated when combining individual PCC estimates through transformed Z-scores. These observations are likely to have an even higher impact on larger studies with more patients. Furthermore, we exclusively analysed immobilized patients. In this light, no conclusion can be drawn about the possible use of ClearSight^®^ in mobile patients or in outpatients. Our study provides by design no insights on whether certain risk groups of pregnant women benefit from the use of continuous blood pressure monitoring. Additionally, concerning the positive judgment regarding the early detection of hypotensive episodes, observer biases can be intrinsic to the approach taken in this study. 

## 5. Conclusions

The measurements of mean arterial blood pressure and diastolic blood pressure using ClearSight^®^ are within the acceptable range, according to AAMI, in our limited cohort. Both patients and medical professionals prefer ClearSight^®^ to conventional oscillometric measurement in terms of comfort and handling. In conclusion, ClearSight^®^ has the potential to become the new standard for close blood pressure monitoring and early detection of supine hypotension syndrome during caesarean section with neuraxial anaesthesia. 

## Figures and Tables

**Figure 1 jcm-11-04498-f001:**
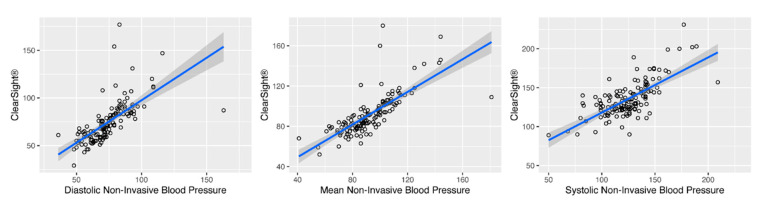
Scatter plots comparing ClearSight^®^ with the reference method for diastolic (**left**, PCC = 0.67), MAP (**centre**, PCC = 0.75) and systolic (**right**, PCC= 0.7) blood pressure. The linear model is depicted as a blue line (regression) with the 95% CI (grey area). *n* = 11 patients with 139 paired measurements.

**Figure 2 jcm-11-04498-f002:**
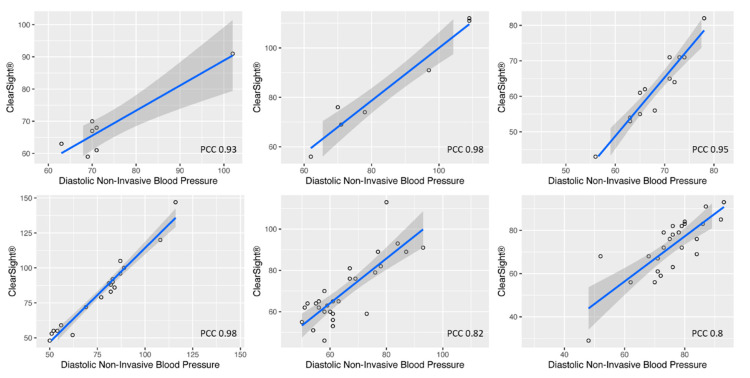
The linear models and Pearson Correlation coefficient based on the paired measurements of six different patients. The linear model is depicted as a blue line (regression) with the 95% CI (grey area). PCC = Pearson correlation coefficient.

**Figure 3 jcm-11-04498-f003:**
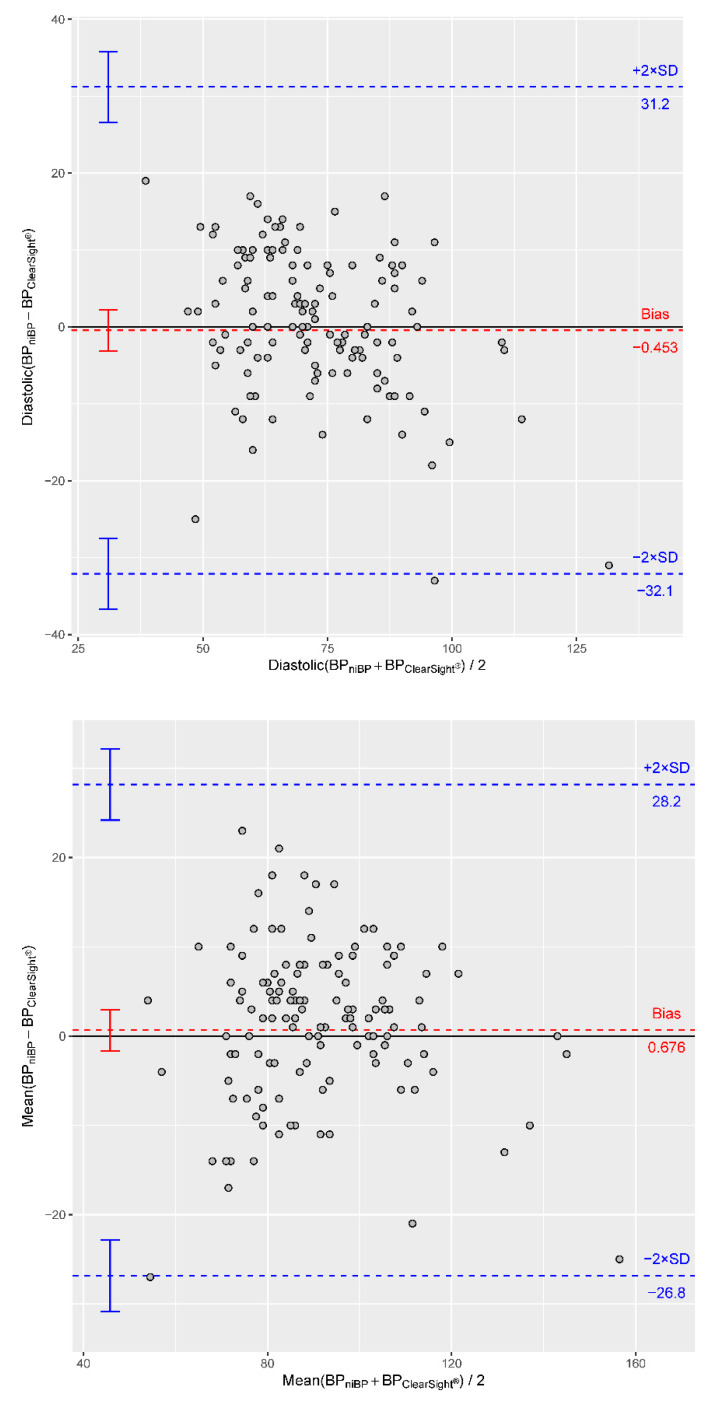
Bland-Altman plots show a bias of −10.8 for systolic, a bias of −0.45 for diastolic and a bias of 0.68 for mean blood pressure measurements between the two benchmarked methods (horizontal dashed lines in red). Additionally, the 2× standard deviation (SD) is provided (horizontal dashed lines in blue). BP_niBP_ = non-invasive blood pressure. BP_clearsight_ = blood pressure measured by Clearsight^®^.

**Table 1 jcm-11-04498-t001:** Patients included in further analysis with the corresponding number of matched measurements.

	Pat.01	Pat.02	Pat.03	Pat.04	Pat.12	Pat.25	Pat.26	Pat.28	Pat.29	Pat.30	Pat.31
Datapoints	13	19	25	30	6	6	7	6	10	7	10

**Table 2 jcm-11-04498-t002:** Medians with interquartile ranges (IQR) for different patient attributes: ASA classification status, age, height, weight and BMI. *n* = 11.

	Median (IQR)
ASA	2 (0)
Age [years]	36 (6.5)
Height [cm]	169 (9)
Weight [kg]	92 (33.3)
BMI [kg/m^2^]	34.21 (17.24)

**Table 3 jcm-11-04498-t003:** Numeric indicators of the Bland–Altman plots.

	Bias	CI of Bias	−2 SD with CI	+2 SD with CI
RRsys	−10.82	−13.8 to −7.8	−46.6 (−51.8 to −41.4)	25.0 (19.8 to 30.2)
RRmean	0.68	−1.6 to 3.0	−26.8 (−30.8 to −22.8)	28.2 (24.2 to 32.2)
RRdia	−0.45	−3.1 to 2.2	−32.1 (−36.7 to −27.5)	31.2 (26.6 to 35.8)

RRsys = systolic blood pressure. RRmean = mean blood pressure. RRdia = diastolic blood pressure. SD = standard deviation. CI = confidence interval.

**Table 4 jcm-11-04498-t004:** Composed results of Bland–Altman analyses for systolic, diastolic and mean blood pressure, from recent studies ordered by the last name of the first author. In addition to the indicators of the corresponding analysis (the observed biases RR and the number of paired measurements), for each study, the year of publication, a description of the cohort and study size is provided.

Author	Year	Cohort	Study Size	RR_sys_ [mmHg]	RR_mean_ [mmHg]	RR_dia_ [mmHg]	Data Points
Lee et al. [25]	2020	Breast cancer surgery	10	−6.0 (−24.3 to 12.3)	−3.8 (−19.7 to 12.1)	−1.2 (−16.1 to 13.7)	245
Noto et al. [26]	2019	Awake carotid endarterectomy	30	−3 (−22.1 to 16)	−6.8 (−20.1 to 6.3)	−9 (−19.7 to 1.5)	2672
Rogge et al. [14]	2019	Obese patients	35	6.8 (−14.4 to 27.9)	1.1 (−13.5 to 15.6)	0.8 (−12.9 to 14.4)	97,623
Sakai et al. [27]	2018	Robot-assisted laparoscopic radical prostatectomy	10	−2.99 (−34.4 to 28.06)	−9.26 (−32.0 to 13.50)	−12.03 (−33.3 to 9.2)	210
Sang-Wook et al. [28]	2021	One-lung ventilation	26	−5.18 (−37.81 to 27.45) ^1^	1.05 (−18.85 to 20.95) ^1^	5.16 (−12.4 to 22.7) ^1^	8408
Schumann et al. [21]	2021	Obese patients	90	−7 (−35 to 20)	−1 (−23 to 21)	0 (−22.0 to 22.0)	538
Tanioku et al. [20]	2020	Cardiovascular surgery	18	13.2 (−21.2 to 47.4)	−3.9 (−19.2 to 11.4)	−9.1 (−23.4 to 5.2)	3068
Yokose et al. [29]	2019	Major abdominal surgery	30	0.7 (−24.6 to 26.0)	7.9 (−7.6 to 21.3)	10.1 (−3.8 to 24.0)	6312

Systolic blood pressure (RRsys); mean blood pressure (RRmean); diastolic blood pressure (RRdia). ^1^ values Bland–Altman: ClearSight minus invasive Measurments.

## Data Availability

Not applicable.

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
