# Peer review of "The Use of Non-Invasive Continuous Blood Pressure Measuring (ClearSight®) during Central Neuraxial Anaesthesia for Caesarean Section—A Retrospective Validation Study"

_jcm, 2022, doi:10.3390/jcm11154498_

Round 1

Reviewer 1 Report

1. The use of noninvasive continuous blood pressure measurement (ClearSight®) during central neuraxial anaesthesia for caesarean section - a retrospective validation study. This is a retrospective study in a series of 31 participants compared conventional BP assessment with non-invasive monitoring using a patented method. The authors report moderate to good correlation with a conventional oscillometric method, perceived ease of use and comfort by providers and participants.

2. The strengths of the manuscript are: how the authors analyzed and summarized prior studies reported the bias and standard differences between the conventional and Clear Sight system for BP surveillance during CD.

The perceived limitations are: this method is not validated in outpatient setting,

The authors need to define what they mean by hypotension, either by defining a set of specific values or clinical signs consistent with the diagnosis.

3. There are multiple previous studies demonstrating the functionality of this method in surgical patients. As a clinical tool, it appears useful as a non-invasive method in immobilized individuals. Use in active individuals who are mobile or the effect of movement on creating artifacts needs additional evaluation, as does the use of this technology in an outpatient setting.

4. The title accurately conveys the message of the paper.

The abstract/summary is a faithful outline of the paper and can be understood without reading the manuscript. No discrepancies exist between the abstract and the remainder of the manuscript.

The introduction succinctly lay the groundwork for what was done and the justification for this study.

Materials and Methods
The hypothesis and the aims are clearly stated.
The study design appropriate to allow the hypothesis to be tested. The methods are clear and this study could be easily reproduced. The data are collected and analyzed properly.

Results
The results are valid based on the methods used. Results are presented as discussed in the methods.

Discussion
The discussion adequately compares and contrast the results with those of other papers that have previously been published.

Study limitations are addressed, specifically the limitations of the technology in routine clinical use.

The conclusion is valid.

Figures and tables are appropriate.

References appropriate, current and comprehensive.

Author Response

Dear Reviewer,

We thank you for this constructive and thorough review of our manuscript. We gladly revised the manuscript extensively and therefore, we did not submit a tracked version. Please find our responses to the comments point by point:

1: We applaud to the reviewer’s excellent summary of our analyses, precisely pinpointing the claims of our manuscript.

2: You are absolutely right that the method investigated has not been validated for the outpatient setting and we introduced in our revised manuscript further clarifications on this point (please see #3).

Since our retrospective study design provided no clear definition, hypotension was diagnosed by the treating anaesthesiologist according to our clinical in-house standards (MAP < 65 mmHg or a systolic blood pressure < 90 mmHg) and treaded with Akrinor. In the revised version of our manuscript, we extended our description of these proceedings:

  • „Hypotension was treated according to established clinical protocols at the discretion of the attending anaesthesiologist“
  • „Hypotension is defined by clinical standards as MAP < 65 mmHg or a systolic blood pressure < 90mmHg.“

3: We were pleased to further clarified your limitations in the Discussion of our revised manuscript - The method studied has not been validated in mobile patients and outpatients:

„In this trial, the measurement accuracy of the ClearSight was investigated exclusively in immobilized patients. Therefore, no conclusion can be drawn about the possible use in mobile patients or outpatients.“

4: We would like to express our warmest thanks to you for the detailed summary and evaluation of the manuscript. We are convinced that our revised version substantially improved in clarity based on your comments.

with best regards

Reviewer 2 Report

With a great interest I have read the work of  Helmer et al on the use of noninvasive continuous blood pressure measurement (ClearSight®) during central neuraxial anesthesia for caesarean section - a retrospective validation study. Authors should be congratulated on a great research question and work delivered.

However, this work faces one of the major methodological limitations– the number of patients. Very small sample (11 analyzed patients) may not lead to secure conclusions. I enjoyed reading your work, and I recommend update of the analysis and publication of much bigger sample of patients. 

Author Response

Dear Reviewer,

First, we would like to thank you for the extraordinary thorough reading of our manuscript and the straightforward expression of appreciation towards the efforts behind our results. You are completely right; the cohort assessed in our study provides no final and definitive answer. We included only about a dozen patients, since each one of them has been subjected to at least five measurements. After thorough quality control, we ended up employing 139 data points for our analyses. In our manuscript, we claim to demonstrate that measurements obtained by the ClearSight® technology correlate with the current clinical gold standard. We are glad to address this point explicitly, since this aspect was discussed in detail before submission of the manuscript:

First, the number of patients included in our validation study is in agreement with the sample size of n>10 patients that is widely employed in complementary validation studies, as reviewed by (DOI 10.1186/s12966-015-0314-1). Second, we calculated ourselves a minimum sample size of n=9.3 according to Cohen’s method, employing parameter values as published before (est. Pearson correlation coefficient PCC=0.78, alpha=0.05, beta=0.8). As a third line of evidence, the Pearson correlations we computed in our analyses are supported by p-values<0.001.

We totally agree with you that it would be desirable to enlarge the sample size to dozens of patients. However, with our study design increasing the sample size would require a re-approval of the ethics amendment (retrospective design), because all available data have been analysed in this manuscript. As a result, the timeline of the current reviewing process would certainly be exceeded. Since all these statistical indicators justify the conclusions we draw from our data, we are convinced that the claims we present in our manuscript are supported well by data from the current cohort size. We further feel that forthcoming studies including significantly more patients will benefit substantially from the insights on the combination of correlation coefficients via Z-score transformations, an approach that to our best knowledge has not been explored so far in aforementioned publications. We were pleased to thoroughly revise the manuscript and in particular to modify the conclusions drawn. The aspect of the limited cohort size was also examined in more detail.

We hope that we will be fulfilling your expectations in the thoroughly revised manuscript and would like to thank you warmly for your effort. 

Reviewer 3 Report

Dear authors,

Thank you for submitting your work. Kindly go through my comments and submit revised version subsequently.

The introduction is very lengthy. The introduction should establish a premise, mention the hypothesis, along with primary and secondary outcomes. Here, it appears like the discussion.

The discussion should start with the mention of key results.

Is ClearSight® US-FDA approved? If no, please mention that.

The conclusion paragraph is very lengthy. It should comprise of 3-5 well-constructed sentences which should summarize the study.

Mention the limitations of this study. It should be in the last paragraph of the discussion, before the conclusions (retrospective study, small sample size, no control group, single-center experience, only obstetric patients included etc.).

Reference 31 (Yang et al) is a paper which has already published an RCT using ClearSight system. What new information is your article going to give, with a retrospective, single-arm study?

Author Response

Dear Reviewer,

we would like to thank you for your very detailed and constructive feedback. We were happy to implement your suggestions concerning the drafting of the manuscript. We hope that we were able to meet your expectations. We gladly revised the manuscript comprehensively. Please find below our point-by-point answers to your questions:

  1. We apologise that our intentions to inform readers in high detail about the various methods of measuring the blood pressure misled us to an inadequately verbose introduction in the manuscript draft. Following your suggestions, we substantially shortened the introduction.

  1. As suggested, we removed an introductory paragraph at the beginning of the discussion section from our manuscript and now discuss the most important results instead. The presentation of our key messages should read much clearer now and we would like to thank you for pointing out this shortcoming.
  2. FDA Approval (Class II) of the ClearSight Platform is filed under the 510(k) Accession Number K182245, with additional information provided through the official FDA portal at the webpage:

https://www.accessdata.fda.gov/scripts/cdrh/cfdocs/cfpmn/pmn.cfm?ID=K182245

Complementing the accuracy tests conducted for the FDA approval of ClearSight, our validation study stands out providing evidence on the application of the device in the preselected cohort. This is in line with recommendations by recent updates (05/2020) to the “medical device regulation” (MDR), which explicitly motivates validation studies and encourages such attempts also after the approval of a device.

  1. Thank you very much for your suggestion to shorten the Conclusion. Following your suggestion, we reduced in the revised version of our manuscript the verbosity of our conclusions, reducing the corresponding section to the key messages by removing the sentences:

- “ClearSight® is an interesting and promising alternative to oscillometric blood pressure measurement for close monitoring in patients undergoing caesarean section with central neuraxial anaesthesia.“

- Data analysis should refrain from pooling data in favour of considering interindividual differences (with compute weighted average Z-score), as this may negatively affect data analysis.

- “Due to the different challenges of conventional continuous measurement methods, this new non-invasive method for continuous blood pressure measurement…“

  1. We were happy to explain the limitations of our work in the last section of the discussion. We agree with your suggestions, although in our opinion the focus on the studied patient cohort (only obstetric patients) does not represent a limitation, as we explicitly refer throughout the manuscript only to a possible use during central neuraxial anaesthesia for caesarean section:

Limitations: “Regarding the interpretation of our results, some constraints may arise due to the retrospective and single-center design of our study allowing us to investigate only a limited number of cases. However, the number of paired measurements allowed to conclude on a significant correlation between ClearSight® and the reference method. Importantly, we demonstrate that in our data set the linear models regressed for the paired measurements of each patient individually differ in slope and shift. Hence, a straightforward analysis on measurements obtained from different patients underestimates the true correlation, but biases can successfully be alleviated when combining individual PCC estimates by transformed Z-scores. These observations are likely to have an even higher impact on larger studies, including more patients. Furthermore, we analysed exclusively immobilized patients. In this light, no conclusion can be drawn about the possible use of ClearSight® in mobile patients or in outpatients. Our study provides by design no insights on whether certain risk groups of pregnant women benefit from the use of continuous blood pressure monitoring. Also, concerning the positive judgment on an early detection of hypotensive episodes, observer biases can be intrinsic to the approach taken in this study. However, in our analysis we calculate linear regressions and correlation coefficients for each patient individually, and we show that results substantially improve when these results are combined via Z-score transformations. As the variability in larger and/or multi-centric studies can be assumed to be higher than in the cohort we study here, these studies will benefit even more from our insights.”

  1. The reviewer is correct that reference #31 of our manuscript describes a study employing ClearSight® in a broader (n=71), randomised, interventional study. In this study, Yang et al. investigated, as primary endpoint, the effect of goal-directed fluid therapy (GDFT) on the incidence of post spinal hypotension. As secondary endpoints, they investigated the absolute injected volume, the dosages of vasopressors, the change in hemodynamic parameters and the maternal/neonatal adverse effects.

However, as a ground-breaking difference to the study we present in our manuscript, Yang et al. did not question the accuracy of ClearSight® measurements in any way. Therefore measurements of a proven and established “gold standard” method are absent from the referenced study. In contrast, one of the main focus of our investigations has been to evaluate the degree and mode of applicability of the relatively novel ClearSight® technology to replace standard monitoring of blood pressure measurement. To this end, we compared ClearSight® measurements to established non-invasive blood pressure measuring. Paramount, we pinpoint that a straightforward approach of comparing ClearSight® to reference measurements from different patients is biased -- most strikingly for diastolic measurements (0.67 vs. 0.9) -- because linear regressions differ in each patient, likely as a result of complex physiological and medical circumstances. Assessing correlation of these measurements altogether therefore is comparing apples to oranges. Therefore unbiased estimates can only be obtained through Z-score transformation of the single correlation coefficients, a well-established technique in the mathematical realm.

To the best of our knowledge, no study so far demonstrated the accuracy of ClearSight® measurements up to the level of detail we present in our manuscript. We are further convinced that our observations on pitfalls in assessing the accuracy of medical devices across multiple patients will be path breaking for forthcoming studies along the same lines.

Round 2

Reviewer 2 Report

Dear Authors,

Thank you for your explanation. Please provide all stated data/information in this review reply as limitations of your study. Furthermore, please report in the methods section the methods and results used for the sample size calculation (as reported in your second paragraph of reply).

Thank you.

Author Response

Dear Reviewer,

We appreciate the very prompt feedback and the possibility to extend the deadline until today. We will gladly add the mentioned limitations to the discussion:

„Due to the limited sample size, we cannot confidently exclude the possibility of under powering some of the single patient analyses.“

In addition, we have also added a slightly modified case number estimation to the methods section:

„In order to confirm the statistical power of our sample size, we employed the function r.test() from the R package “pwr”: for the minimal observed PCC of 0.73, a power of 0.8 and a significance level of 0.05, we thus obtained a minimal sample size of n=11.“

Appreciating your great intrest and useful commentaries on our manuscript, we think that we have implemented all requested changes to your satisfaction.

With best regards

Reviewer 3 Report

Dear authors,

Thank you for revising the manuscript based on the suggestions of the reviewers. 

Author Response

We would like to thank you for the effort, time and helpful comments. The constructive feedback has certainly contributed elemantarily to the improvement of the manuscript. We will be happy to edit the wording of the manuscript again.